# Participation of the ABC Transporter *CDR1* in Azole Resistance of *Candida lusitaniae*

**DOI:** 10.3390/jof7090760

**Published:** 2021-09-15

**Authors:** Valentin Borgeat, Danielle Brandalise, Frédéric Grenouillet, Dominique Sanglard

**Affiliations:** 1Institute of Microbiology, University of Lausanne and University Hospital, CH-1011 Lausanne, Switzerland; valentin.borgeat@unil.ch (V.B.); danielle.brandalise@chuv.ch (D.B.); 2Pole de Biologie Médicale, Centre Hospitalier Universitaire (CHU) Besançon, F-25000 Besançon, France; fgrenouillet@chu-besancon.fr; 3Chrono-Environnement Unité Mixte de Recherche (UMR) 6249, Centre National de la Recherche Scientifique (CNRS), Université Bourgogne Franche-Comté, F-25000 Besançon, France

**Keywords:** ABC transporter, drug resistance, *Candida*

## Abstract

*Candida lusitaniae* is an opportunistic pathogen in humans that causes infrequent but difficult-to-treat diseases. Antifungal drugs are used in the clinic to treat *C. lusitaniae* infections, however, this fungus can rapidly acquire antifungal resistance to all known antifungal drugs (multidrug resistance). *C. lusitaniae* acquires azole resistance by gain-of-function (GOF) mutations in the transcriptional regulator *MRR1*. *MRR1* controls the expression of a major facilitator transporter (*MFS7*) that is important for fluconazole resistance. Here, we addressed the role of the ATP Binding Cassette (ABC) transporter *CDR1* as additional mediator of azole resistance in *C. lusitaniae. CDR1* expression in isolates with GOF *MRR1* mutations was higher compared to wild types, which suggests that *CDR1* is an additional (direct or indirect) target of *MRR1*. *CDR1* deletion in the azole-resistant isolate P3 (V688G GOF) revealed that MICs of long-tailed azoles, itraconazole and posaconazole, were decreased compared to P3, which is consistent with the role of this ABC transporter in the efflux of these azoles. Fluconazole MIC was only decreased when *CDR1* was deleted in the background of an *mfs7*Δ mutant from P3, which underpins the dominant role of *MFS7* in the resistance of the short-tailed azole fluconazole. With R6G efflux readout as Cdr1 efflux capacity, our data showed that R6G efflux was increased in P3 compared to an azole-susceptible wild type parent, and diminished to background levels in mutant strains lacking *CDR1*. Milbemycin oxim A3, a known inhibitor of fungal ABC transporters, mimicked efflux phenotypes of *cdr1*Δ mutants. We therefore provided evidence that *CDR1* is an additional mediator of azole resistance in *C. lusitaniae,* and that *CDR1* regulation is dependent on *MRR1* and associated GOF mutations.

## 1. Introduction

*Candida* spp. can cause invasive and opportunistic fungal diseases associated with variable mortality in immunocompromised patients as well as those with cancer and under hematopoietic cell transplantation [1]. *Candida albicans* is considered to be the most common species isolated from blood cultures, however, fungal infections caused by non-albicans *Candida* species are on the rise [2]. Among the non-albicans *Candida* species, *Candida lusitaniae* is an uncommon pathogen that accounts for approximately 1% of isolates in large datasets of adult and pediatric patients with candidemia and other forms of invasive candidiasis [3]. *C. lusitaniae* has the ability to rapidly acquire resistance to currently used antifungals including azoles, echinocandins, polyenes and pyrimidine analogs. While resistance to single agents has been reported in this species [4,5], resistance to multiple agents (multidrug resistance, MDR) has also been documented [6,7]. *Candida auris*, which is phylogenetically closely related to *C. lusitaniae* and is a recently emerging fungal pathogen, exhibits quite similar characteristics in terms of rapid drug resistance acquisition and multidrug resistance [8].

Antifungal drug resistance occurs by several molecular mechanisms involving specific genome mutations [9]. In *C. albicans*, at least two transcriptional regulators involved in drug resistance, *TAC1* and *MRR1,* have been described. While *TAC1* regulates ABC transporter genes such as *CDR1* and *CDR2*, *MRR1* regulates transporter genes of another family called major facilitators, among which is *MDR1* [10,11]. In *C. lusitaniae*, an important mediator of fluconazole resistance is the transcription factor *MRR1* (CLUG_00542). Mutations in this factor (gain-of-function mutations, GOF) result in the transcriptional activation of several genes, the Major Facilitator transporter *MFS7* (CLUG_01938, also named *MDR1*, 56% identity with *MDR1* of *C. albicans*) being one of these [5,7,12]. The deletion of both genes in *C. lusitaniae* is critical for azole resistance [5,7,12]. Interestingly, *MRR1* and its target *MFS7* are also involved in resistance against the pyrimidine analog 5-fluorocytosine (5-FC), probably by active efflux of the drug [7].

Transcriptional studies have revealed other potential mediators of fluconazole resistance in *C. lusitaniae* [5,7,12]. Inspecting the genes that were upregulated in azole-resistant isolates or those upregulated by the presence of *MRR1* GOF mutations in *C. lusitaniae*, one interesting candidate was the homolog of the *C. albicans* ABC transporter *CDR1* named CLUG_03113. This gene (about 90% similarity with *CDR1* of *C. albicans*) was renamed *CDR1* in the present study. Here, we addressed the role of *CDR1* in the azole resistance of *C. lusitaniae*. We confirmed first that a novel *MRR1* GOF from a clinical azole-resistant isolate was associated with *CDR1* upregulation. We observed that *CDR1* expression was under the control of *MRR1.* The inactivation of *CDR1* in *C. lusitaniae* resulted in the decreased efflux capacity of the fluorophore rhodamine 6G, and was associated with decreased resistance against long-tailed azoles such as itraconazole and posaconazole, which are known ABC transporter substrates. Together our data highlight a novel regulatory association between the ABC transporter *CDR1* and the transcriptional activator *MRR1* in *C. lusitaniae*.

## 2. Materials and Methods

### 2.1. Strains, Media and Primers

*C. lusitaniae* isolates were grown in complete medium Yeast Extract Peptone Dextrose (YEPD) (1% Bacto peptone, Difco Laboratories, Basel, Switzerland), 0.5% yeast extract (Difco) and 2% glucose (Fluka, Buchs, Switzerland) at 30 °C under agitation. YEPD agar plate were used containing 2% agar (Difco). When required, YEPD was supplemented with 200 µg/mL nourseothricin (Werner BioAgents, Jena, Germany) or with 250 µg/mL hygromycin (Mediatech, Manassas, VA, USA). Strains and primers used in this study can be found in Appendix A. As earlier described, P3 isolate was recovered from a patient treated with multiple agents about 2 months after recovery of the initial susceptible isolate P1 [6].

### 2.2. Case Report of C. lusitaniae Infection

A female preterm neonate was born at 25.7 weeks of gestational age by spontaneous delivery after premature rupture of membranes. She was admitted to the Neonatal Intensive Care Unit (NICU, CHRU Besançon) and weighed 880 g. She was intubated at birth for poor respiratory effort. Apgar scores were 3 and 5 at 1 and 5 min, respectively. Central venous catheters were placed. *Proteus mirabilis* (>100 CFU, Colony Forming Unit) and *C. lusitaniae* (10 CFU) were isolated from vagina of the mother before delivery. *Proteus mirabilis* was also isolated in gastric aspirate of infant, and diagnosis of chorioamnionitis was retained. Cefotaxime and amikacine were administered intravenously for 8 and 2 days, respectively. Intravenously, fluconazole prophylaxis was a standard protocol in the NICU for premature infants with a birth weight of less than 1000 g, or a gestational age of less than 28 weeks, in agreement with international guidelines [13,14]. On day 2, fluconazole was given at a dose of 3 mg per kilogram of body weight every third day for the first two weeks. Sampling for fungal colonization detection (axillae, umbilicus, anus) allowed repeated isolation of few *C. lusitaniae* on axillae (day 2: 10 CFU, day 12: 40 CFU). Urine sampled on day 14 showed 700 CFU/mL *C. lusitaniae.* Fluconazole regimen were thus increased to 6 mg/kg every two days until day 24, and later continued at prophylactic dosage 3 mg/kg every three day until day 43. Infant received concomitantly new cure of cefotaxime (from day 29 to day 36) for pneumonia due to *Haemophilus parainfluenzae.* Systematic fungal screening (groin, axillae, nose, urine) performed on day 40 showed high colonization of groin with *C. lusitaniae* (>200 CFU, isolate later called DSY4941). E-test method revealed high minimal inhibitory concentration (MIC) to fluconazole (128 µg/mL; read at 48 h). On day 42, infant presented with fever of unexplained origin, leading to fluconazole withdrawal and liposomal amphotericin B implementation (5 mg/kg/d, from day 42 to day 55). Blood cultures remained negative. She was extubated at day 50. No other infectious complication was further suspected. She was discharged from NICU, then hospital, at day 63 and day 91, respectively. Long-term follow-up did not show any abnormality or neurodevelopmental impairment. *C. lusitaniae* isolated from infant before day 40 and from mother’s vagina were unfortunately not stored for further investigations.

### 2.3. Deletion of CDR1

In order to inactivate *CDR1* (CLUG_03113) in different strain backgrounds, a CRISPR approach was used. First, a repair fragment was constructed by fusion PCR using the hygromycin resistance marker from pYM70 [15] and *C. lusitaniae* genomic DNA. The first PCR fragment amplified the 5′ UTR (untranslated region) of *CDR1* from isolate P1 with primers ClCDR1-P1 and ClCDR1-TEF-5R (20-bp overlap with pYM70). The second PCR fragment amplified the selective marker *HygR* from pYM70 with primers CLCDR1-TEF-5f (20-bp overlap with *CDR1*) and ClCDR1-ACT1-3r (20-bp overlap with pYM70). The third PCR fragment amplified the 3′ UTR of *CDR1* with primers ClCDR1-P2 and ClCDR1-ACT1-3f (20-bp overlap with *CDR1*). These PCRs were carried out with in a peqStar Instrument (Peqlab, Erlangen, Germany) with Taq DNA Polymerase (New England Biolabs, Ipswich, MA, USA). PCR fragments were purified with NucleoSpin Gel and PCR clean-up kit (Macherey-Nagel, Oensingen, Switzerland). The final PCR was performed with the three purified fragments and Phusion DNA Polymerase (New England Biolabs, Ipswich, MA, USA) in the presence of the nested primers ClCDR1-P3 and ClCDR1-P4 with the addition of 1 M betaine, which increases PCR yields by inhibiting secondary structure formation [16].

To target deletion of *CDR1*, the RNA-protein complexes (RNPs) approach was used that employs reconstituted purified Cas9 protein in complex with scaffold and gene-specific guide RNAs [17]. gRNAs specific for *CDR1* (crRNAs: ClCDR1_5_Cas9 and ClCDR1_3_Cas9) were obtained from IDT (Integrated DNA Technologies, Inc., Coralville, IO, USA) as CRISPR guide RNA (crRNA), which contains 20-bp homologous to the target gene fused to the scaffold sequence. Gene-specific RNA guides were designed in silico using Geneious Prime (Biomatters, Ltd., Auckland, New Zealand). RNPs were created as following: briefly, crRNAs and tracrRNA (a universal transactivating CRISPR RNA) were dissolved in RNase-free distilled water (dH_2_O) at 100 µM and stored at −80 °C. Before use, each crRNA and the tracrRNA were diluted to 16 µM and the complete guide RNAs were generated by mixing equimolar concentrations of each crRNA and tracrRNA to obtain 8 µM solutions. These mixes were incubated at 95 °C for 5 min and cooled down to room temperature. The Cas9 nuclease 3NLS (60 µM stock from IDT) was diluted to 8 µM in dH_2_O. RNPs were assembled by mixing each complete guide RNA (3.6 µL of each gene-specific crRNA/tracrRNA) with 3 µL of diluted Cas9 protein, followed by incubation at room temperature for 5 min. Transformation of *C. lusitaniae* cells was carried out by electroporation with 3.3 µL of each gene-specific RNPs (6.6 µL total), 40 µL of *C. lusitaniae* cells and 1–2 µg of repair construct (up to 3.4 µL volume) as earlier described [7]. Selection of transformants was performed on hygromycin-containing agar plates after 1–2 days of incubation at 30 °C. Correct *CDR1* deletion was verified by PCR with primer pairs ClCDR1-P1/Hygro_1949R and ClCDR1-3-verif/ACT1-pYM70 to obtain expected amplification fragments.

### 2.4. Restoration of MRR1 Alleles

In order to restore the *MRR1* GOF mutation (V654A) of isolate DSY4941 in the background of a *mrr1*Δ isolate (DSY5416), a CRISPR approach was used as published [7]. An *MRR1* template was first constructed which contained the GOF mutation, which was achieved by fusion PCR of two different fragments that used pDS2140 (containing the wild type *MRR1* allele flanked by an *SAT1* marker) [7]. The amplifications used primer pairs pDS1918-P1/V654A-R and V654A-F/ClMRR1-SacI with Phusion DNA Polymerase (New England Biolabs, Ipswich, MA, USA) in the presence of 1 M betaine. The resulting fragments were pooled in a final PCR using the nested primer pair ClMRR1-Apa/MRR1-3_rev_new with the same above-described conditions. The final PCR fragment was used to transform DSY5416 (P3 *mrr1*Δ) using the published CRISPR-based procedure [7]. Transformants were selected on nourseothricin agar plates and correct isolates exhibiting the *MRR1* GOF mutation V654A were selected upon PCR verification with primer pair ClMRR1_F/Cl_MRR1_3377_R followed by Sanger sequencing.

### 2.5. Rhodamine 6G Efflux

Efflux of rhodamine 6G (R6G) was carried out as described [18]. Briefly, cells were grown overnight in YEPD and were diluted in 5 mL YEPD to grow at 30 °C under constant agitation until a density of 2 × 10^7^ cells/mL was obtained. Cells were centrifuged, washed, and resuspended in 2 mL PBS (pH 7). Energy deprivation was next achieved by 1 h incubation in PBS at 30 °C. R6G was then added at a concentration of 10 µg/mL and the incubation was continued for 1 h. After this incubation time, cells were centrifuged, washed with PBS at 4 °C, and resuspended in a final volume of 200 µL PBS. Fifty microliters of individual strains were diluted in 50 µL PBS and aliquoted in a 96-well microtiter plate, which was placed in a LUMIstar Omega microplate reader (BMG LABTECH, Ortenberg, Germany) with temperature control at 30 °C. Baseline fluorescence emission (excitation wavelength: 340 nm; emission wavelength: 555 nm) was recorded as relative fluorescence units (RFU) for 5 min and glucose (1% final concentration) was next injected to initiate R6G efflux. As a control, no glucose was added to separate aliquots of each strain. Data points were recorded in duplicates for 60 min at 1 min intervals and were plotted in Graph Prism software (Version 9.1.0, GraphPad Software, San Diego, CA, USA).

### 2.6. qPCR Assays

Total RNA was extracted from log phase cultures grown in YEPD at 30 °C under constant agitation as described [7]. Gene expression levels were determined by real-time quantitative reverse transcription-PCR (qRT-PCR) in a StepOne real-time PCR system (Applied Biosystems, Foster City, CA, USA). cDNA was prepared with a PrimeScript RT reagent kit (Perfect Real Time) (Takara). Subsequent qPCRs were performed with a 0.2 µM concentration of each primer and a 0.1 µM concentration of TaqMan probes (see Appendix A) and iTaq Supermix with ROX (Amine-reactive carboxy-x-rhodamine) (Bio-Rad, Reinach, Switzerland) according to the manufacturer’s instructions. Assays were performed in biological triplicates and normalized to *ACT1* (CLUG_03241).

### 2.7. Minimum Inhibitory Concentrations (MIC) Assays

To determine MICs of *C. lusitaniae* isolates to antifungal agents, we used Sensititre YeastOne (Thermo Fisher Scientific, Reinach, Switzerland) 96-wells plates. Overnight cultures were diluted in 11 mL of YeastOne inoculum broth (Thermo Fisher Scientific, Reinach, Switzerland) to reach a final concentration of 5 × 10^3^ cells/mL. Next, 100 µL of the inoculum was added in each well of the plates. The results were read after 24 h incubation at 35 °C. Serial dilution assays were performed as described earlier [7].

### 2.8. Sequencing

Sanger sequencing was carried out by Microsynth AG (Balgach, Switzerland).

## 3. Results

### 3.1. CDR1 Expression Patterns in C. lusitaniae

Previous studies performed in *C. lusitaniae* with azole-resistant isolates have reported that *CDR1* expression is increased when compared to wild type isolates or isolates lacking *MRR1* (Table 1). Isolates P3, P4 and U04 contain each of the *MRR1* GOF mutations (V668G in P3 and P4; Y813C in U04). The presence of these mutations triggers not only the upregulation of *MFS7* but also of *CDR1*. While screening a collection of recovered *C. lusitaniae* isolates from a French hospital, we recovered a specific azole-resistant isolate (DSY4941; MICs in Appendix A) from an infected premature child (case report in Material and Methods). This isolate exhibited a novel *MRR1* mutation (V654A). In order to evaluate the effect of this mutation not only on *MFS7* but also on *CDR1* expression, it was introduced on a wild type *MRR1* allele. The results in Figure 1 show that the V654A GOF (DSY5639) elevated both *MFS7* and *CDR1* expression compared to a wild type *MRR1* allele. These elevated expression levels were similar to those obtained with the known V668G mutation (DSY5439). Both GOF mutations resulted in an increase in azoles and 5-FC MICs compared to the wild type *MRR1* (Appendix A). Data in Figure 1 also confirms that in the absence of *MRR1* (DSY5416, P3 *mrr1*Δ), both *CDR1* and *MFS7* were expressed at low levels compared to GOF isolates.

These data were confirmed by probing the expression levels of *CDR1* and *MFS7* in isolate P3 and its derived *MRR1* mutant (DSY5416, P3 *mrr1*Δ): both *CDR1* and *MFS7* expression were decreased by about 10-fold when *MRR1* was deleted in this isolate background (Figure 2). Expression of *CDR1* and *MFS7* in the *MRR1* mutant from the parent azole-susceptible isolate P1 (DSY5658, P1 *mrr1*Δ) did not vary as extensively when compared to P1 (Figure 2). Taken together, our results, and those from others, illustrate that *CDR1* expression is under the control, either directly or indirectly, of at least *MRR1*.

### 3.2. Effect of CDR1 on Antifungal Susceptibility

While it is known that *MFS7* participates not only in fluconazole but also in 5-FC resistance of *C. lusitaniae* [7], the contribution of *CDR1* in these phenotypes is not yet clear. *CDR1* was, therefore, deleted in the background of different strain backgrounds and mutants (Appendix A). We observed that in the absence of *CDR1* in the isolate P3 (P3-*cdr1*Δ), the MICs of the long-tailed azoles itraconazole and posaconazole were decreased by 8-fold while not altering the MIC of the short-tailed azole fluconazole (Figure 3 and Table 2). Moreover, 5-FC resistance of isolate P3 was not dependent on *CDR1* as judged by the measured 5-FC MICs (Table 2). These trends were similar when *CDR1* was deleted in isolates complemented with *MRR1* GOF alleles (Appendix A). In the background of an *mrr1*Δ mutant in P3, *CDR1* deletion has little additional effect on azole MICs, probably due to the presence of *MFS7* and the low *CDR1* expression of the *mrr1*Δ mutant. Serial dilutions on YEPD agar containing itraconazole or fluconazole confirmed these observations: deletion of *CDR1* in isolate P3 decreased resistance to itraconazole but not to fluconazole. Deletion of *MRR1* in P3 decreased resistance to both itraconazole or fluconazole as a consequence of decreased expression of both *MFS7* and *CDR1* (Figure 4).

As expected, the deletion of *MFS7* in P3 decreased fluconazole MIC by 8-fold. However, the additional *CDR1* deletion in the *mfs7*Δ mutant decreased fluconazole MIC by 256-fold compared to the initial isolate P3 (Figure 3 and Table 2). The effect of *CDR1* on fluconazole MIC is, therefore, better distinguished in the background of an *mfs7*Δ mutant of isolate P3 since *MFS7* is the major contributor of fluconazole resistance in *C. lusitaniae* [7]. These data also suggest that *CDR1* covers a broad range of azoles, including short- and long-tailed azoles, which is known from other *CDR1*-like ABC transporters of other *Candida* spp. [19,20].

*CDR1* was also deleted in the background of the azole-susceptible isolate P1, which is the parent of P3. In this isolate background, the effect of *CDR1* deletion was again best observed in an *mfs7*Δ mutant, in which the fluconazole MIC was decreased by at least 4-fold compared to P1 (Table 3 and Appendix A).

### 3.3. Probing Cdr1 Activity by R6G Efflux

We next explored the efflux activity of Cdr1 in *C. lusitaniae* by using an efflux reporter system based on the fluorescence measurements of rhodamine 6G (R6G) with whole cells. This system was used previously to measure efflux activities of ABC transporters in *C. albicans* and *C. glabrata* [18,21]. As shown in Figure 5, R6G efflux activity was higher in P3 compared to P1 after glucose was added in the incubation buffer. Maximal slope in R6G efflux kinetics was 2.7-fold increased in P3 compared to P1, which is consistent with the higher expression of *CDR1* in P3 compared to P1. Efflux activity was strongly diminished after glucose addition when *CDR1* was deleted in the background of both isolates P3 and P1. We confirmed that R6G efflux was principally mediated by Cdr1 since *MFS7* deletion in both P1 and P3 isolates did not alter efflux profiles compared to wild types. Interestingly, when *MRR1* was deleted in P3, efflux activity after glucose supply was similar to isolate P1 (Figure 6). Given that *CDR1* expression is down-regulated in *mrr1*Δ mutants compared to *MRR1* GOF isolates (Figure 2), this result is consistent with the assumption that Cdr1 is responsible for R6G efflux in *C. lusitaniae*. The combined deletion of *MRR1* and *CDR1* in both P1 and P3 backgrounds and the associated decrease in R6G efflux to basal levels confirmed this hypothesis.

The importance of Cdr1 for efflux activity was also addressed by efflux inhibition with milbemycin oxim (AOx3), a known ABC transporter inhibitor in other *Candida* spp. [21]. The addition of AOx3 to P1 and P3 isolates reduced R6G efflux to basal activities (Figure 7). When the same experiments were carried out with *cdr1*Δ mutants from the same isolates, we noticed that AOx3 was still able to slightly decrease R6G efflux compared to mutant cells incubated with glucose only (Figure 7). This suggests the presence of additional transporter activity potentially mediated by other ABC transporters in this fungal species. Interestingly, in the presence of AOx3, the MICs of the long-tailed azoles posaconazole and itraconazole were strongly reduced in isolate P3, while MICs of the short-tailed azoles fluconazole and voriconazole were little affected (Table 4). This tendency was confirmed in isolate P1 for itraconazole MICs. These results are consistent with the AOx3-dependent inhibition of ABC transporters, which are known to exhibit efflux activity on long-tailed azoles [22]. The MICs of the short-tailed azoles fluconazole and voriconazole in isolate P3 are mostly attributed to *MFS7*, a major facilitator that is upregulated in P3. Major facilitators are not inhibited by milbemycins [23,24,25], which, therefore, preserved the short-tailed azole MICs of isolate P3 even in the presence of AOx3.

## 4. Discussion

Antifungal resistance is mediated by several mechanisms involving transcriptional changes, genome mutations and post-transcriptional modifications [9]. The upregulation of multidrug transporters and the resulting drug resistance is a very widespread mechanism in fungal pathogens [26]. In *C. albicans*, upregulation of ABC transporters and major facilitators results in resistance to azoles [22]. The increased expression of these transporters compared to drug-susceptible isolates relies on the transcriptional control of two major regulators, *TAC1* and *MRR1* [10,11]. In *C. albicans*, *TAC1* regulates the ABC transporter genes *CDR1* and *CDR2*, while *MRR1* regulates the major facilitator *MDR1* [10,18,27,28]. In the past few years, *TAC1* and *MRR1* homologs have been described in several other fungal pathogens including *Candida parapsilosis*, *Candida tropicalis* and *C. auris* [29,30,31]. A *TAC1* homolog exists in *C. lusitaniae* (CLUG_02369) with about 55% similarity to the *C. albicans TAC1*. This gene has not been associated, up to now, with mutations in azole-resistant isolates (upregulating *MFS7* and *CDR1*) compared to susceptible isolates [5,7,12]. Therefore, the role of CLUG_02369 in drug resistance is likely not to be relevant, although cannot be excluded. In *C. auris*, it has been established that *TAC1b* (a *TAC1* homolog) can regulate the expression of at least *CDR1* [32]. Among the three *MRR1* homologs of *C. auris*, only the deletion of *MRR1a* resulted in decreased azole susceptibility of an isolate of Clade III [33]. The full repertoire of *MRR1a* targets in *C. auris* remains unknown still. In *C. lusitaniae*, only one *MRR1* homolog has been investigated for its role in azole resistance [7,12]. *MRR1* of *C. lusitaniae* regulates at least the major facilitator *MFS7*, an important mediator of fluconazole and 5-FC resistance in this species [7]. Here, we show with several approaches that *CDR1* of *C. lusitaniae* may be considered as another direct or indirect target of *MRR1*. This is supported by (i) the upregulation of *CDR1* in the presence of *MRR1* GOF mutations; (ii) the loss of *CDR1* upregulation when *MRR1* is deleted. This is an unexpected result given that *CDR1*-like genes in other *Candida* spp. have been shown to be under the control of *TAC1* homologs [32,34]. The *C. lusitaniae MRR1* may have, therefore, accommodated different targets compared to other *Candida* spp. Transcriptional rewiring is a common feature of fungal pathogens [35]. Future studies should address the *MRR1* regulon in *C. lusitaniae* by more elaborate approaches through promoter occupancy experiments.

Cdr1 activity was measured here by R6G efflux, which was conveniently recorded by a whole cell-based assay over time. R6G is considered as a specific substrate for ABC transporters [21,36]. This was indirectly confirmed here in efflux assays with transporter mutants. While R6G efflux activity was decreased in mutant P3 *cdr1*Δ compared to the parent P3, efflux activity was not changed in the P3 *msf7*Δ mutant, thus suggesting that Cdr1 was the major mediator of R6G efflux in *C. lusitaniae*. The *C. lusitaniae* genome contains several other ABC transporters [7,37], and it is possible that these transporters could also contribute to R6G efflux activity. The data presented in Figure 5 and Figure 7 are consistent with this hypothesis. *CDR1* deletion in isolate P3 as well as milbemycin efflux inhibition could not reduce R6G efflux to background activity (e.g., efflux activity in absence of glucose), thus suggesting the participation of additional transporters. The *C. lusitaniae* genome contains a close relative to *CDR1* (CLUG_03113), CLUG_02179 (or EJF14_60115 in isolate P1) with 95% similarity. This is reminiscent of the *CDR1*/*CDR2* pair of ABC transporters in *C. albicans* [38]. These two transporters are co-regulated by *TAC1* [10]. When both *CDR1* and *CDR2* are deleted in *C. albicans*, this results in increased azole susceptibility compared to a wild type and single deletion mutants [38]. However, EJF14_60115 is not regulated by *MRR1* GOF mutations in *C. lusitaniae* [7,12] and, therefore, is not likely to participate in azole resistance of this fungal pathogen. Nevertheless, it is possible that EJF14_60115 could still contribute to the basal R6G activity that is detectable in *cdr1*Δ mutants. Future studies could address this possibility.

## 5. Conclusions

In conclusion, we showed here that the expression of the *C. lusitaniae* ABC transporter *CDR1* is regulated by *MRR1* and is participating in the resistance to long-tailed azoles. It is remarkable that *C. lusitaniae MRR1* is able to regulate simultaneously two different types of multidrug transporters (ABC transporter and major facilitator) that are known to play major roles in the acquisition of antifungal resistance. This allows *C. lusitaniae* to enlarge its repertoire of defenses against drug stresses. This way, the combined *CDR1* and *MFS7* upregulation triggered by GOF mutations in *MRR1* confer not only decreased susceptibility to long-tailed azoles but also to short-tailed azoles. Since *MRR1* GOF mutations may arise even in azole-untreated patients, this regulatory relationship should not be underestimated in antifungal therapy.

## Figures and Tables

**Figure 1 jof-07-00760-f001:**
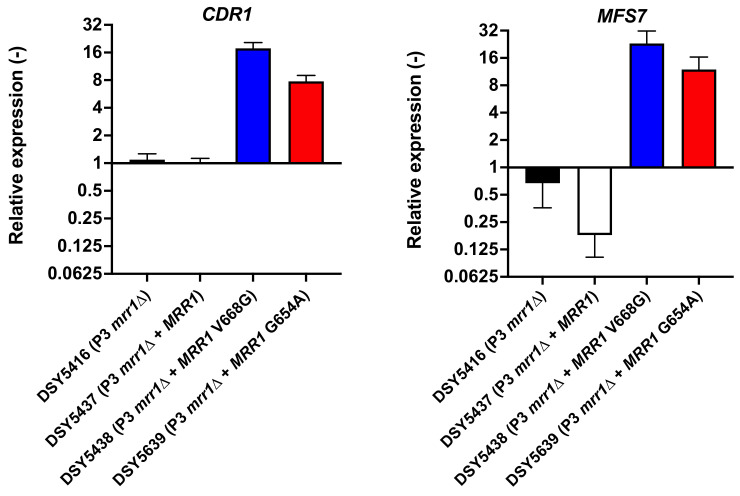
*MRR1* GOF alleles trigger the upregulation of multidrug transporters *CDR1* and *MFS7*. qPCRs were performed as described in Material and Methods. Expression fold-changes were reported to the parent isolate DSY5416 for *CDR1* and *MFS7* expression patterns. Blue and red colors indicate the effect of *MRR1* GOF mutations V668G and G654A, respectively.

**Figure 2 jof-07-00760-f002:**
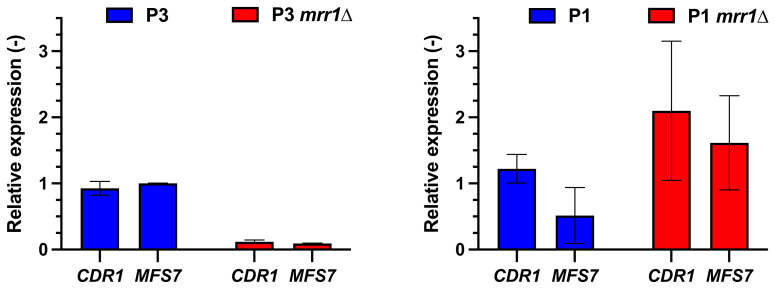
*CDR1* and *MSF7* are regulated by *MRR1* in isolate P3. qPCRs were performed as described in Material and Methods. Expression fold-changes were reported to the parent isolates P1 and P3, respectively. The P3 isolate contains the *MRR1* GOF mutation V668G, while P1 contains a wild type *MRR1* allele.

**Figure 3 jof-07-00760-f003:**
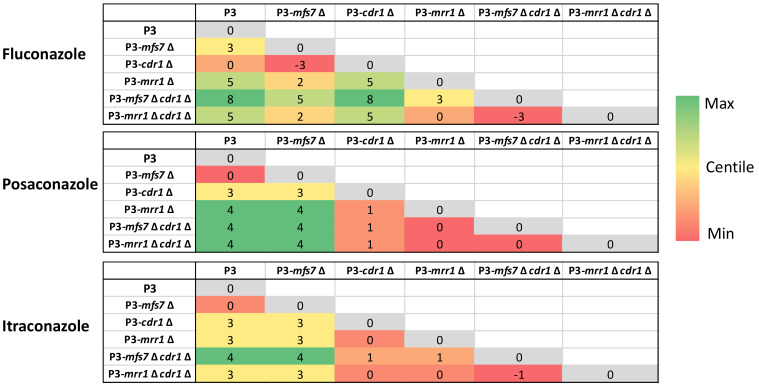
Pairwise comparisons of MICs between isolate P3 and derived mutants. Fold-changes are given in log2 scale. Comparisons were made always starting from top row. Values > 0 indicate MIC decreases, while values <0 indicate MIC increases. Greyed colored boxes indicate comparisons between same isolates (Value = 0).

**Figure 4 jof-07-00760-f004:**
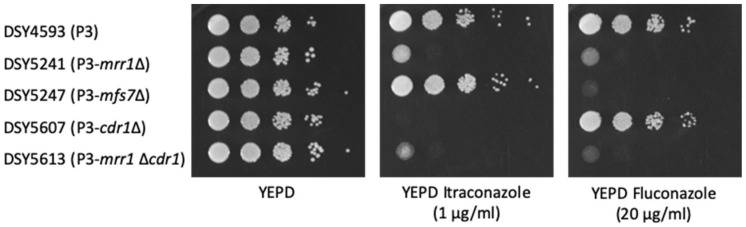
Serial dilution assays for itraconazole and fluconazole. Isolates were 10-fold serially diluted and spotted in the corresponding agar plates starting from approximately 10^7^ cells/mL from inoculum cultures. Plates were incubated at 35 °C for 24 h.

**Figure 5 jof-07-00760-f005:**
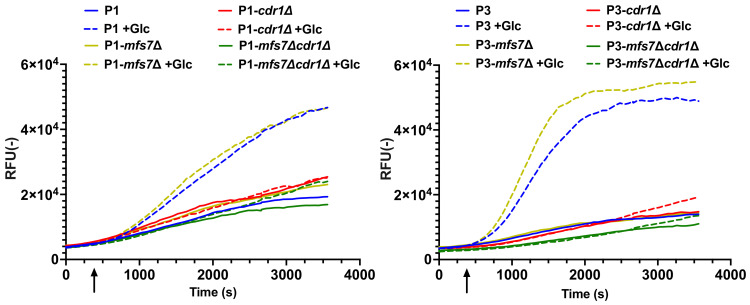
Rhodamine 6G efflux activities are dependent on the presence of *CDR1*. Efflux activities were performed as described in Material and Methods. Means of duplicates are reported. Glucose was added 300 s after start of measurements (arrow). Maximal slopes were calculated with Graph Prism. Maximal slopes were derived from data points ranging between 740 and 1040 s and calculated by linear regression implemented in Graph Prism.

**Figure 6 jof-07-00760-f006:**
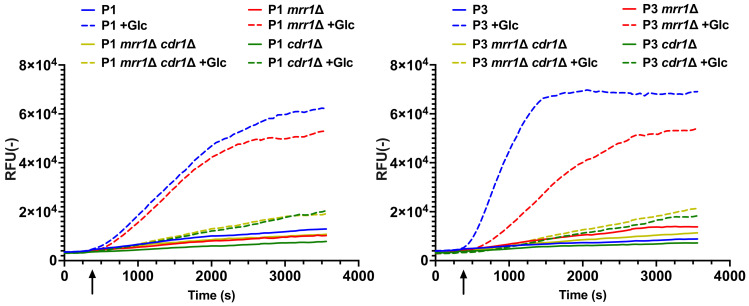
Rhodamine 6G efflux activities are dependent on the presence of *CDR1* and *MRR1*. Efflux activities were performed as described in Material and Methods. Means of duplicates are reported. Glucose was added 300 s after start of measurements (arrow).

**Figure 7 jof-07-00760-f007:**
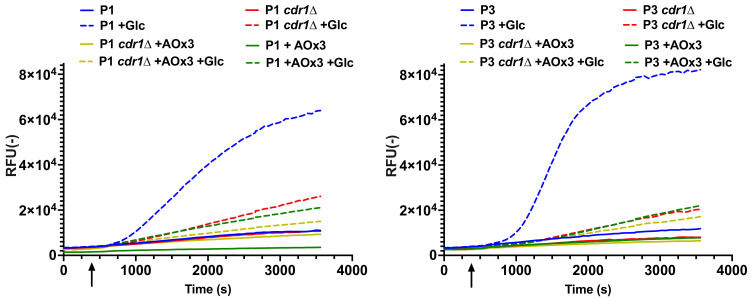
Milbemycin oxim 3 (AOx3) inhibits rhodamine 6G efflux. Efflux activities were performed as described in Material and Methods. Means of duplicates are reported. Glucose was added 300 s after start of measurements (arrow). Milbemycin AOx3 (5 µg/mL) was added to indicated samples at start of experiment.

**Table 1 jof-07-00760-t001:** *CDR1* expression in different isolate backgrounds.

Isolate	Fold-Change (vs. Azole-Susceptible Isolates)	Experimental Condition	Reference
DSY4593 (P3)	2,4	RNAseq	[7]
DSY4661 (P4)	4,7	RNAseq	[7]
U04	2,6 ^a^	RNAseq	[12]
DSY4593 (P3)	5,0	qPCR	[6]
DSY4661 (P4)	3,5	qPCR	[6]

^a^ fold-change was calculated by comparison with an *mrr1*Δ mutant from U04.

**Table 2 jof-07-00760-t002:** Antifungal MICs of *C. lusitaniae* P3 and mutant derivatives.

Antifungal Drugs	MIC (µg/mL)					
	P3	P3-*mfs7*Δ	P3-*cdr1*Δ	P3-*mrr1*Δ	P3-*mfs7*Δ*cdr1*Δ	P3-*mrr1*Δ*cdr1*Δ
Anidulafungin	0.12	0.25	0.12	0.12	0.12	0.12
Micafungin	0.12	0.12	0.12	0.06	0.06	0.03
Caspofungin	0.12	0.25	0.12	0.12	0.12	0.12
5-Flucytosine	64	4	32	0.5	0.5	0.5
Posaconazole	0.25	0.25	0.03	0.015	0.015	0.015
Voriconazole	0.5	0.12	0.25	0.015	<0.008	0.015
Itraconazole	0.5	0.5	0.06	0.06	0.03	0.06
Fluconazole	32	4	32	1	<0.12	1
Amphotericin B	0.5	0.5	0.5	0.25	0.25	0.25

**Table 3 jof-07-00760-t003:** Antifungal MICs of *C. lusitaniae* P1 and mutant derivatives.

Antifungal Drugs	MIC (µg/mL)					
	P1	P1-*mfs7*Δ	P1-*cdr1*Δ	P1-*mrr1*Δ	P1-*mfs7*Δ*cdr1*Δ	P1-*mrr1*Δ*cdr1*Δ
Anidulafungin	0.12	0.06	0.12	0.12	0.06	0.12
Micafungin	0.06	0.06	0.06	0.03	0.12	0.06
Caspofungin	0.12	0.25	0.12	0.12	0.25	0.12
5-Flucytosine	4	0.5	4	1	0.5	1
Posaconazole	0.03	0.015	0.03	0.015	0.015	0.015
Voriconazole	<0.008	<0.008	<0.008	<0.008	<0.008	<0.008
Itraconazole	0.06	0.06	0.06	0.06	0.03	<0.008
Fluconazole	0.5	0.25	0.25	1	<0.12	1
Amphotericin B	0.25	0.5	0.25	0.25	0.25	0.25

**Table 4 jof-07-00760-t004:** Effect of milbemycin AOx3 (5 µg/mL) on antifungal MICs.

Antifungal Drugs	MIC (µg/mL)			
	P1	P1-AOx3	P3	P3-AOx3
Anidulafungin	0.12	0.06	0.12	0.06
Micafungin	0.06	0.06	0.06	0.06
Caspofungin	0.12	0.25	0.12	0.06
5-Flucytosine	4	0.5	64	64
Posaconazole	0.03	0.015	0.25	<0.008
Voriconazole	<0.008	<0.008	0.5	0.25
Itraconazole	0.06	0.06	0.5	<0.008
Fluconazole	0.5	0.25	32	16
Amphotericin B	0.25	0.5	0.5	0.5

## Data Availability

The datasets generated during and/or analyzed during the current study are available from the corresponding author upon request.

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
