# Peer review of "Participation of the ABC Transporter CDR1 in Azole Resistance of Candida lusitaniae"

_jof, 2021, doi:10.3390/jof7090760_

Round 1

Reviewer 1 Report

This manuscript by Borgeat et al describes an interesting connection between CDR1 mediated azole resistance and the MRR1 transcriptional regulator in Candida lusitaniae. By examining a series of patient isolates, the authors identify a new gain of function mutation in MRR1 (V654A) and show that it leads to increased expression of both MFS7 and CDR1. Targeted gene deletions are used to examine the role for these genes in resistance, by constructing mutants in the background of patient isolates with or without the MRR1 GOF mutation.

The authors present clear data that CDR1 appears to be regulated by MRR1, whereas they note in other species it is regulated by TAC1 homologs. What is the evidence that TAC1 is also playing a role- have homologs been identified and tested in C. lusitaniae, or is there evidence from the comparative genomics of how this regulation may have diverged between species? 

A general comment is that the gene names and the mapping of genes between species needs to be clarified. At line 60, the text states that the CLUG_03113 was renamed CDR1 for "practical purposes", however it is unclear what these criteria were and how MFS7 is related to MDR1 in other species. At line 39e, a "close relative to CDR1" is noted, however it is unclear if this is a close relative of C. albicans CDR1 or another gene in C. lusitanae termed CDR1. Please consider adding the CLUG ids for all the major genes that were studied to make this clear to readers, and explain the context of sequence comparisons (level of identity in alignment or phylogenetic evidence of orthology vs paralogy for example).

Can the authors comment how how CDR1 levels compare to MDR1 in terms of transcript abundance rather than analyzing each separately using relative expression? 

Minor comments

The case report is described extensively and is extraordinary for the treatment of this patient. It would be helpful to clarify at what point during the course of treatment that the P1 and P3 samples were collected. 

What do the colors signify in Figure 1? If they do not serve a specific purpose it would be more clear to make this a B/W image.

Line 240-241- Expression of both CDR1 and MFS7 appear to increase when MRR1 is deleted in the P1 background. What explains this- are these significant differences?

Figure 3- It is confusing to have the same comparisons displayed in inverse on both halves of this chart (comparing A to B on top, and B to A on the bottom with different colors). Please consider only showing one half of these plots

Figures 5-6-The tests of efflux with fluorescent probes are interesting, however would be more direct if fluor labeled azoles had been tested.

Author Response

Reviewer 1

We thank reviewer 1 for his constructive comments.

  • The authors present clear data that CDR1 appears to be regulated by MRR1, whereas they note in other species it is regulated by TAC1 homologs. What is the evidence that TAC1 is also playing a role- have homologs been identified and tested in C. lusitaniae, or is there evidence from the comparative genomics of how this regulation may have diverged between species? 

Response to 1: A TAC1 homolog (CLUG_02369) with about 55% similarity the C. albicans TAC1 exists in C. lusitaniae. This gene has up to now not been associated with SNPs in azole-resistant isolates (upregulating MFS7 and CDR1) compared to susceptible isolates of C. lusitaniae (Kannan et al. and Demers et al). TAC1 (CLUG_02369) has not been yet inactivated in C. lusitaniae to appreciate its effect on the expression of multidrug transporters and azole resistance. However, given the lack of SNP occurrences in this gene in existing azole-resistant isolates, our hypothesis is that the role TAC1 (CLUG_02369) in drug resistance of C. lusitaniae is not prominent. We have integrated these discussion points in the revised version of the manuscript

(lane 372-376): A TAC1 homolog exists in C. lusitaniae (CLUG_02369) with about 55% similarity the C. albicans TAC1. This gene has not been associated up to now with mutations in azole-resistant isolates (upregulating MFS7 and CDR1) compared to susceptible isolates [5,7,12]. Therefore, the role of CLUG_02369 in drug resistance is likely not relevant, although not excluded

With regards to comparative genomics and divergence of gene regulation, it is difficult to make predictions that will rule several species, given that transcriptional rewiring is known to operate between distinct species. The accepted paradigm stipulates that TAC1 homologs regulate ABC-transporters, while MRR1 homologs regulate Major Facilitators. While this holds true in C. albicans, this transcriptional pattern has not been systematically interrogated in other Candida species yet.

  • A general comment is that the gene names and the mapping of genes between species needs to be clarified. At line 60, the text states that the CLUG_03113 was renamed CDR1 for "practical purposes", however it is unclear what these criteria were and how MFS7 is related to MDR1 in other species. At line 39e, a "close relative to CDR1" is noted, however it is unclear if this is a close relative of C. albicans CDR1 or another gene in C. lusitaniae termed CDR1. Please consider adding the CLUG ids for all the major genes that were studied to make this clear to readers, and explain the context of sequence comparisons (level of identity in alignment or phylogenetic evidence of orthology vs paralogy for example).

Response to 2: We have modified the sentence introducing CLUG_03113 and naming as “CDR1”. We have also added locus names of each gene discussed in this paper (MFS7 and ACT1) to avoid confusion. In the revised version:

(lane 58-61): Inspecting the genes that were upregulated in azole-resistant isolates or those upregulated by the presence of MRR1 GOF mutations in C. lusitaniae, one interesting candidate was the homolog of the C. albicans ABC-transporter CDR1 named CLUG_03113. This gene (about 90% similarity with CDR1 of C. albicans) was renamed CDR1 in the present study.

  • Can the authors comment how how CDR1 levels compare to MDR1 in terms of transcript abundance rather than analyzing each separately using relative expression? 

Response to 3: It is not possible to compare transcript abundance of MFS7 and CDR1 with the qPCR probes used here. However, transcript abundance can be extracted from previously published RNAseq data (Kannan et al). In this work, the RPMK (Reads Per Kilobase of transcript per Million reads mapped) is about 2-3 fold higher for MFS7 than it is for CDR1 in the azole-resistant isolate P3. This suggests that transcript abundance is slightly higher for MFS7 than it is for CDR1.

Minor comments

  • The case report is described extensively and is extraordinary for the treatment of this patient. It would be helpful to clarify at what point during the course of treatment that the P1 and P3 samples were collected.

Response to 4: As published in Asner et al (2015), the time elapsed between isolation of P1 and P3 was about 2 months. The patient was under fluconazole therapy when P3 was isolated, which corresponded to the azole resistance profile of this isolate. In the revised version:

(Lane 78-80): As earlier described, P3 isolate was recovered from a patient treated with multiple agents about 2 months after recovery of the initial susceptible isolate P1 [6].

  • What do the colors signify in Figure 1? If they do not serve a specific purpose it would be more clear to make this a B/W image.

Response to 5: Explanations were now given in legend of Figure 3: “Blue and red colors indicate the effect of MRR1 GOF mutations V668G and G654A, respectively”.

  • Line 240-241- Expression of both CDR1 and MFS7 appear to increase when MRR1 is deleted in the P1 background. What explains this- are these significant differences?

Response to 6: Deletion of MRR1 in the azole-susceptible isolate increases the expression of CDR1 and MFS7, as noted by the reviewer, however the increase was not significant. This effect was also observed in an independent study in which MRR1 was also deleted in another azole-susceptible isolate (Demers et al.). MFS7 (called MDR1 in the Demers study) was upregulated in the mrr1 mutant compared to wild type, however, the upregulation did not reach the level of expression obtained in the azole-resistant isolate. There are yet no clear explanations for this behavior.

  • Figure 3- It is confusing to have the same comparisons displayed in inverse on both halves of this chart (comparing A to B on top, and B to A on the bottom with different colors). Please consider only showing one half of these plots

Response to 7: This is now corrected in the revised manuscript.

  • Figures 5-6-The tests of efflux with fluorescent probes are interesting, however would be more direct if fluor labeled azoles had been tested.

Response to 8: We understand that azoles with fluorophores could now be used to address their efflux in fungal cells. However we do not have access to these labelled substances. In addition, the labelled substances are more adequate for subcellular localization studies and intracellular content. They have not been used to measure effluxed substances outside of cells.

Reviewer 2 Report

In the manuscript “Participation of the ABC-transporter CDR1 in azole resistance of Candida lusitaniae“, the authors performed the set of experiments, proving that C. lusitaniae Cdr1 transporter is mainly responsible for the pathogen resistance to the long-tailed azoles. In contrast to other well studied Candida species, the CDR1 gene is under control of Mrr1 transcription factor, which controls also expression of  the MDR1 gene, encoding MFS7 efflux transporter responsible for the resistance to short-tailed azoles. Moreover, C. lusitaniae clinical isolates acquire spontaneously GOF-type mutations in MRR1, slipping away from the azole treatment.

In my opinion, presented work is interesting and it is important to the other fungal researchers, working on Candida species. The carefully thought-out experiments gave the interesting results, although I see few points concerning this work, which makes it unclear, as listed below:

First of all: according to the phylogenetic analysis of C. lusitaniae, formerly known as Clavispora lusitaniae, it belongs to the Metschnikowiaceae clade, contrary to C. albicans (Debaryomycetaceae). Hence, the comparison of two different species and the regulation of some genes, in my opinion, is pointless. It is better to describe a novel type of regulation by Mrr1 in the genus Clavispora, rather than to compare them to C. albicans. Moreover, in the yeasts, the nomenclature of the gene or protein names is well established (according to CGD, gene CDR1, protein Cdr1 or Cdr1p). Please, consider updating the nomenclature and taxonomy in the manuscript.

Row 47: While TAC1 regulates...expression of the genes encoding ABC-transporters, of course? Please, revise the entire manuscript for confounding terms, concerning relations between genes and proteins.

Row 109: please, give some coordinates to the CDR1 gene, i.e. locus name of the closest orthologue, because some C. lusitaniae strains (e.g. FDAARGOS_655) possesses two ORFs with the same protein name.

Row 123: what was the reason to use betaine? Please, add citation

Row 145: PCR and Southern analysis would be useful to prove proper deletion

Row 207: Have the authors checked double V668G-Y813C mutant? Does two mutations suppress each other the phenotype?

Figure 2 in the legend P3 mrr1delta is the deletion of the mutated allele? It should be signed somehow

Row 252: resistance of C. lusitaniae – citation needed

Row 379: the alignment of the promoter regions of C. lusitaniae CDR1 with the promoters of the C. albicans and C. auris CDR1 would be helpful. Then the indication of the putative Mrr1-binding sites, absent in other Candida species, would clarify presented results

Author Response

We thank reviewer 2 for his constructive comments.

  • First of all: according to the phylogenetic analysis of lusitaniae, formerly known as Clavisporalusitaniae, it belongs to the Metschnikowiaceae clade, contrary to C. albicans(Debaryomycetaceae). Hence, the comparison of two different species and the regulation of some genes, in my opinion, is pointless. It is better to describe a novel type of regulation by Mrr1 in the genus Clavispora, rather than to compare them to C. albicans.

Response to 1: We understand the arguments of the reviewer. We have used the C. albicans knowledge on TAC1 and MRR1 as a discussion comparison. We however addressed the current knowledge on TAC1 and MRR1 with C. auris, which is more closely related to C. lusitaniae than is C. albicans. In any case, MRR1 regulation of CDR1 is a novel finding, independent on C. albicans and C. auris knowledge.

  • Moreover, in the yeasts, the nomenclature of the gene or protein names is well established (according to CGD, geneCDR1, protein Cdr1 or Cdr1p). Please, consider updating the nomenclature and taxonomy in the manuscript.

Response to 2: We have corrected the gene/protein nomenclature whenever necessary.

  • Row 47: While TAC1 regulates...expression of the genes encoding ABC-transporters, of course? Please, revise the entire manuscript for confounding terms, concerning relations between genes and proteins.

Response to 3: We have corrected the gene/protein nomenclature whenever necessary. For example in revised manuscript:

Lane 47-49: While TAC1 regulates ABC-transporters genes such as CDR1 and CDR2, MRR1 regulates transporters genes of another family called Major Facilitators and among which MDR1 [10,11].

  • Row 109: please, give some coordinates to theCDR1 gene, i.e. locus name of the closest orthologue, because some  lusitaniae strains (e.g. FDAARGOS_655) possesses two ORFs with the same protein name.

Response to 4: Gene locus name for CDR1 (CLUG_03113) was given.

  • Row 123: what was the reason to use betaine? Please, add citation

Response to 5: Betaine is known PCR yields and also to better resolve GC-rich regions during amplification procedures.  We added a sentence in h revised version of the manuscript and a reference.

(lane 124-127): The final PCR was performed with the three purified fragments and Phusion DNA Polymerase (New England Biolabs, Ipswich, MA, USA) in the presence of the nested primers ClCDR1-P3 and ClCDR1-P4 with the addition of 1 M betaine, which increases PCR yields by inhibiting secondary structure formation [16].

16: Jensen, M.A.; Fukushima, M.; Davis, R.W. DMSO and Betaine Greatly Improve Amplification of GC-Rich Constructs in De Novo Synthesis. Plos One 2010, 5, e11024, doi:10.1371/journal.pone.0011024.

  • Row 145: PCR and Southern analysis would be useful to prove proper deletion

Response to 6: PCR verifications of mutant genotypes is a current method for mutant verifications and we provided details on the chosen approach in the Material and Methods section. Southern approaches are currently not carried out. We however recently genome sequenced mfs7/cdr1 mutants and the data showed the expected absence of the deleted genes. Such data can be provided on request, however cannot be integrated in the current manuscript since they belong to another study.

  • Row 207: Have the authors checked double V668G-Y813C mutant? Does two mutations suppress each other the phenotype?

Response to 7: We did not carry out the construction of a double GOF (V668G-Y813C) in the current study, since the inclusion of such construction was not necessary for the message of the paper. It has been reported that MRR1 can harbor loss of function mutations, which neutralize the effect of GOF mutations (Demers). The two mutations V668G and Y813C appear in separate MRR1 alleles and are associated with GOF phenotypes. Combining the two mutations has therefore little interest. Combining the two mutations could rather result in stronger GOF phenotypes in the opinion of the authors.

  • Figure 2 in the legend P3 mrr1delta is the deletion of the mutated allele? It should be signed somehow

Response to 8: Yes the P3 mrr1delta correspond to the deletion of the mutant allele. The text of the legend was modified as requested.

(lane 238-239): The P3 isolate contains the MRR1 GOF mutation V668G, while P1 contains a wild type MRR1 allele.

  • Row 252: resistance of lusitaniae – citation needed

Response to 9: citation added

  • Row 379: the alignment of the promoter regions of lusitaniae CDR1 with the promoters of the C. albicans and C. auris CDR1 would be helpful. Then the indication of the putative Mrr1-binding sites, absent in other Candida species, would clarify presented results

Response to 10: We don’t think that promoter alignments could be helpful to help explanations supporting the regulation of CDR1 from C. lusitaniae by MRR1, given the genetic distance between the different species. The idea of adding Mrr1 binding sites on the CDR1 promoters is certainly interesting, however the consensus of MRR1 binding is only known for C. albicans and may change in other Candida spp. Considering the binding site consensus “DCSGHD” as published by Shubert et al (AAC, 2011), it was found in the C. lusitaniae CDR1 promoter in multiple copies. This in silico analysis is however not sufficient to conclude that Mrr1 binds the CDR1 promoter in C. lusitaniae. Rather, in vivo occupancy of Mrr1 should be carried out in C. lusitaniae itself. Such experiments are now underway however cannot be integrated in the current manuscript since not yet concluded.

This manuscript is a resubmission of an earlier submission. The following is a list of the peer review reports and author responses from that submission.